

# Jet aircraft lubrication oil droplets as contrail ice-forming particles

Joel Ponsonby[1], Leon King[2], Benjamin J. Murray[2] and Marc E. J. Stettler[1*]

[1]Department of Civil and Environmental Engineering, Imperial College London, London, SW7 2AZ, United Kingdom
[2]School of Earth and Environment, University of Leeds, Woodhouse, Leeds, LS2 9JT, United Kingdom

*Correspondence to*: Marc Stettler (m.stettler@imperial.ac.uk)

**Abstract**. The radiative characteristics and lifetime of contrails are dependent on the number concentration of ice-forming particles in the engine exhaust plume. Aircraft gas turbine engines produce a variety of particles, yet it is understood that non-volatile black carbon aggregates are the dominant source of ice-forming particles with typical, fossil-derived jet fuel. However, with cleaner combustion technologies and the adoption of alternative fuels (e.g., hydrogen or synthetic aviation fuel), non-

volatile black carbon particle emissions are expected to decrease or even be eliminated. Under these conditions, contrail properties will depend upon the concentration and characteristics of particles other than black carbon. Ultrafine (<100 nm) jet lubrication oil droplets constitute a significant fraction of the total organic particulate matter released by aircraft, however their ability to form contrail ice crystals is hitherto unexplored. In this work, we experimentally investigate the activation and freezing behaviour of lubrication oil droplets using an expansion chamber, assessing their potential as ice-forming particles.

We generate lubrication oil droplets with a geometric mean mobility diameter of $(100.9 \pm 0.6)$ nm and show that these activate to form water droplets despite their hydrophobicity. These subsequently freeze when the temperature is below ~235 K. We find that nucleation on lubrication oil droplets should be considered in future computational studies - particularly under soot-poor conditions - and that these studies would benefit from particle size distribution measurements at cruise altitude. Overall, taking steps to reduce lubrication oil number emissions would help reduce the climate impact of contrail cirrus.

**1. Introduction**

Condensation trails (contrails) are line-shaped ice clouds that are produced by aircraft (Kärcher, 2018). If the ambient air is supersaturated with respect to ice, contrails can persist and spread, forming contrail-cirrus, which has been estimated to cover ~0.1% of the world in cloud (Sausen et al., 1997; Burkhardt and Kärcher, 2009). The distribution of contrail-cirrus coverage depends sensitively upon ambient conditions (Gierens et al., 2020), aircraft/fuel properties (Jeßberger et al., 2013; Schumann,

2000; Bier and Burkhardt, 2019), flight time (Teoh et al., 2022) and season (Stuber et al., 2006). Contrail-cirrus interacts with both solar and terrestrial radiation, resulting in a net warming effect when averaged globally (Burkhardt and Kärcher, 2011). Due to its warming potential, contrail-cirrus has been estimated to account for approximately two-thirds of effective radiative forcing by the present-day aviation industry, the remainder of which is due to accumulated $CO_2$ and emissions of nitrogen oxides (Lee et al., 2021).



Contrails form by a combination of two components: water vapour and aerosol particles. These undergo turbulent mixing within the aircraft plume to generate contrail ice crystals (Bier et al., 2022), which are produced by the mechanism of condensation followed by homogeneous freezing (Kärcher et al., 2015). Firstly, water vapour condenses on aerosol particles entrained within the exhaust plume. These aerosol particles serve as condensation nuclei (CN) and activate into water droplets, which continue to grow as the plume cools. Activated CN subsequently freeze homogeneously at ~235 K, generating local ice

crystal concentrations of up to $10^4$ cm$^{-3}$ (Schumann and Heymsfield, 2017) that are visible as bright white streaks behind cruising aircraft. The water supersaturation and temperature at which these phase transitions occur depend upon the physicochemical nature of the CN and are dictated by the cooling conditions within the plume (Kärcher et al., 1996). Given that contrail ice formation relies upon the availability and properties of CN, it is important to understand the nature of the CN that are present in aircraft exhaust plumes.

Aircraft exhaust plumes contain three principal aerosol components: ambient particles, non-volatile particulate matter (nvPM) and volatile particulate matter (vPM). Ambient tropospheric aerosol includes mineral dust, black and organic carbon, other organic material and sulfates (Kärcher et al., 2007); these are entrained within aircraft exhaust emissions at effective concentrations of $10^{12}$ - $10^{13}$ particles per kg of fuel burned (Kärcher and Yu, 2009). nvPM is defined as material that does not volatilize when heated to 623 K (350 °C) (Saffaripour et al., 2020). The majority of nvPM produced by aircraft is carbonaceous

black carbon, which is a product of incomplete combustion (Kärcher, 1999). Aircraft-generated black carbon exists in the form of branched, fractal aggregates, which comprise many agglomerated primary particles (Petzold et al., 1999). Turbulent mixing within an aircraft engine leads to a diversity of particle growth history, which makes it challenging to define a single primary or aggregate mean diameter (Vander Wal et al., 2022). Instead, geometric mean diameters for aggregates and primary particles have been reported to fall within the ranges 10 - 50 nm and 12 - 26 nm, respectively (Saffaripour et al., 2020). In contrast to

nvPM, vPM forms from condensable vapours in the aircraft exhaust after they have been sufficiently cooled. vPM comprises an organic fraction deriving from unburned fuel and aircraft lubrication oil, and an inorganic fraction mainly consisting of sulfates, which form by oxidation of fuel sulfur compounds (via $SO_2$ and $SO_3$) (Timko et al., 2010). Downstream of the engine exit plane, vPM can either condense upon nvPM or nucleate new particles, a preference which has been shown to depend upon the engine thrust (Timko et al., 2010; Yu et al., 2019). Measurements at ground level show that for low engine thrust settings

(~7%), sulfuric acid is critical for initiating new particle formation; nucleation-mode particles (~20 nm) are produced with high organic content, which increases total particle numbers by up to two orders of magnitude downstream of the engine exit plane (Timko et al., 2010). At higher engine thrust settings (> 65%), the number of nucleation-mode particles decreases, and the total particle volume distribution is dominated by the nvPM particle mode (50 - 150 nm) (Yu et al., 2019).

    Aircraft burning kerosene-based fuel typically produce between $10^{11}$ and $10^{16}$ nvPM particles per kg of fuel (ICAO:

Aircraft Engine Emissions Databank, 2023); under these "soot-rich" conditions, the emission index of nvPM and the effective emission index of contrail ice crystals increase roughly in proportion (Kärcher and Yu, 2009). Water condensation followed by homogeneous freezing is effectively facilitated by the majority of nvPM particles, with the largest and most hygroscopic particles activating preferentially (Koehler et al., 2009).



In the future, nvPM emissions are projected to decrease as a result of engine emissions regulations and climate mitigation strategies. In 2017, all member states of the International Civil Aviation Organization (ICAO) agreed on a new engine emissions standard effective from 1 January 2023 (ICAO, 2017), setting a thrust-dependent limit on nvPM number emissions during landing and take-off. Further, in 2016 all member states of the ICAO agreed upon a Carbon Offsetting and Reduction Scheme for International Aviation (ICAO, 2019). The aim of this proposal is to achieve carbon neutral growth from 2021, through the targeted use of sustainable aviation fuels (SAF) which have lower life-cycle carbon dioxide emissions compared to fossil-derived jet fuel (Wang and Tao, 2016). Use of blended fuels comprising SAF and kerosene reduce nvPM emissions (Durdina et al., 2021) and subsequent ice crystal numbers by 50% to 70% with respect to pure kerosene (Bräuer et al., 2021; Voigt et al., 2021). Therefore, given that nvPM emissions are projected to decrease, and with the introduction of SAF and other alternative fuels, assessing non-combustion sources of CN and their contrail-forming potential is of paramount importance. Note that adoption of non-fossil fuels (e.g., hydrogen and electric aircraft), would result in drastic nvPM emissions reductions as the fuel contains no carbon.

Of the three aerosol components outlined above, lubrication oil and ambient particles may be considered as potential sources of non-combustion CN. Indeed, using a contrail parcel model, researchers have demonstrated that in the absence of nvPM (under "soot-poor" conditions) condensation and freezing occurs competitively between condensed vPM and ambient particles (Kärcher and Yu, 2009). In the literature, the contribution of jet lubrication oil derivatives towards the total organic exhaust fraction is commonly quantified using the ratio of ion fragment intensity at $m/z = 85$ and 71, obtained using mass spectrometry (Yu et al., 2012). This approach has been paired with positive matrix factorization to demonstrate that factors associated with lubrication oil components are uncorrelated with aircraft engine thrust setting and combustion efficiency (i.e., non-combustion sources); these are instead related to the engine design and lubrication oil type (Timko et al., 2014).

Lubrication oil can be released from overboard breather vents (Eastwick et al., 2006; Nie et al., 2018) or via clearance seals (Flitney, 2014), which form part of the aircraft oil recirculation system (Hunecke, 2003). To that end, lubrication oil droplets with volumetric mean diameters in the range 250 - 350 nm have been identified by sampling directly from the breather vents (Yu et al., 2010). Measurements performed by another group, 15 m downstream of an engine exit plane, also found that a significant proportion of lubrication oil existed in the particle size range > 300 nm (Timko et al., 2010). The researchers found that for a different engine, 90% of the condensed vPM mass derived from lubrication oil and was confined to a particle size range 80 - 500 nm, qualifying that the characteristics of lubrication oil emissions are sensitive to engine technology. In a separate study, measurements taken 30 – 150 m from active taxiways also identified lubrication oil contributions towards vPM between 5% and 100% in the particle size range 50 – 700 nm, in association with the nvPM particle mode (Yu et al., 2012).

More recently, it has been suggested that lubrication oil droplets can effectively nucleate from the vapour phase even without a nvPM condensational sink (Ungeheuer et al., 2022). Measurements taken 30 m downstream (of the engine exit plane) of an aircraft operating at 85% thrust have demonstrated that lubrication oil is the dominant contributor towards vPM, particularly in the nucleation mode (< 30 nm) (Yu et al., 2019). This was corroborated by near-runway sampling at Narita and Frankfurt International Airports (Fushimi et al., 2019; Ungeheuer et al., 2020), where researchers found that the majority of





compounds detected in nucleation mode particles (respectively defined as < 30 nm, 10 - 56 nm) could be attributed to jet lubrication oil components.

Therefore, given that: (a) lubrication oil droplets constitute a significant fraction of condensable aircraft vPM, (b) the relative proportion of lubrication oil in aircraft exhaust emissions is projected to increase, due to nvPM emissions reductions and (c) the impact of lubrication oil droplets on contrails is hitherto unexplored, this paper addresses their ability to function as contrail ice-forming particles. First, an experimental setup is designed to investigate both the activation and freezing properties of the droplets. The activation properties of lubrication oil droplets will then be compared to the activation properties

of nvPM, obtained from the literature. Next, the freezing mechanism adopted by the activated lubrication oil droplets will be discussed and the implications placed in the context of contrail formation. Finally, the paper will conclude with a comment on the significance of ultrafine lubrication oil droplets as a source of contrail ice-forming particles.

## 2. Theory

Contrails form when a plume of hot, moist aircraft exhaust gas mixes with cool, dry ambient air at constant pressure (Kärcher,

2012). The temperature and water vapour partial pressure ($T_M$, $p_{w,M}$) of the mixing contrail air parcel can be determined at any point during its evolution, provided the exhaust ($T_E$, $p_{w,E}$) and ambient ($T_A$, $p_{w,A}$) conditions are known (Yau and Rogers R.R., 1989). The contrail mixing line can be approximated to be linear between the exhaust and ambient boundary conditions. As such, this behaviour can be alternatively described using a single boundary condition (i.e., $T_A$, $p_{w,A}$) and the gradient of the mixing line, $G$, which is related to measurable aircraft and fuel properties by (Schumann, 1996),

$$\frac{dp_{w,M}}{dT_M} = G = \frac{c_p EI_w P_T}{\varepsilon Q(1-\eta)},$$     (1)

where $P_T$ is the total air pressure; $c_p$ is the specific enthalpy of the exhaust gas; $EI_w$ is the emission index of water, the mass of water released per mass of fuel burned; $Q$ is the heat released per mass of fuel burned; $(1 - \eta)$ is the fraction of heat transferred to the exhaust gas and $\varepsilon$ is the ratio of gas constants for water vapour and dry air.





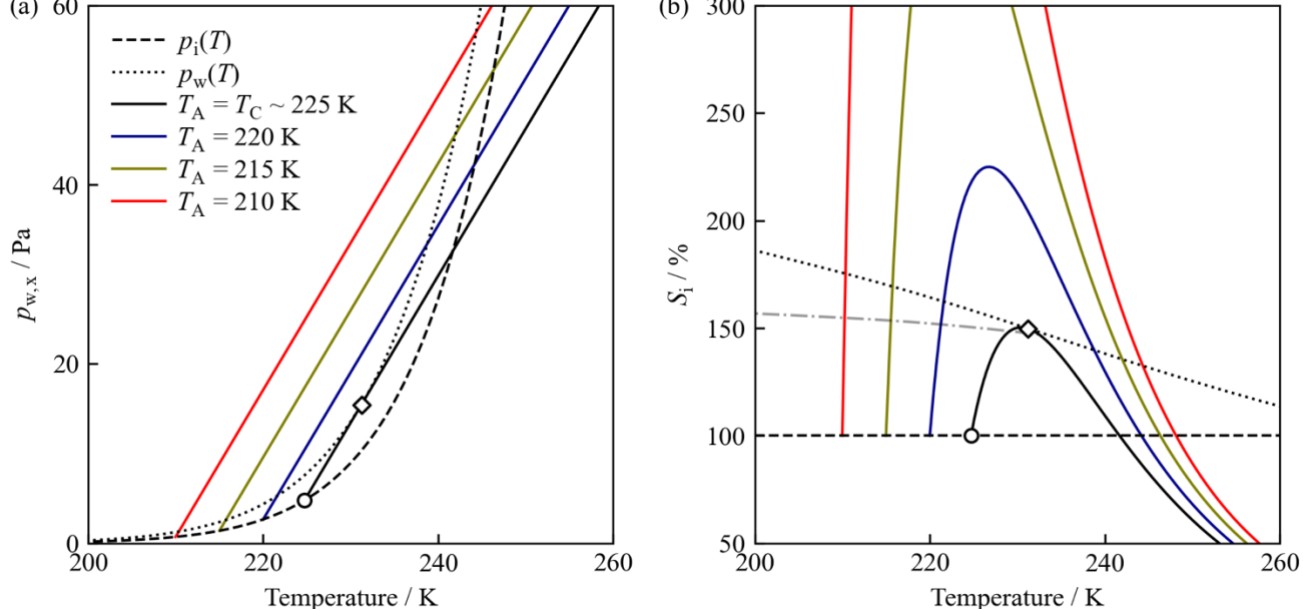

**Figure 1: Contrail mixing. (a) Dotted and dashed black lines trace the saturation vapour pressures above supercooled water and ice respectively. Linear contrail mixing lines are described by $G = 1.64$ and terminate at $p_{w,A} = p_{ice}$. These are plotted for $T_A$ values of 210 K, 215 K and 220 K. An additional mixing line is drawn that makes tangential contact with $p_{liq}$ (position of the diamond marker) at a threshold ambient temperature, $T_C \sim 225$ K. (b) Water vapour partial pressures have been normalised by $p_{ice}$ so that the ordinate represents the saturation ratio, $S_i$. The "Koop line" is also plotted (grey dot-dashed line) (Koop et al., 2000).**

Contrail mixing behaviour is presented in Fig. 1a along with curves that trace the saturation vapour pressures above ice ($p_{ice}$) and supercooled water ($p_{liq}$) (Murphy and Koop, 2005). Each contrail mixing line is described by $G = 1.64$, corresponding to typical aircraft and fuel properties ($P_T$, $EI_w$, $Q$, $\eta$) of (250 mb, 1.23 kg.kg$^{-1}$, 43.2 MJ, 0.3) (Kärcher et al., 2015) and terminates at $p_{w,A} = p_{ice}$. For the grey mixing lines, $T_A$ takes values of 210 K, 215 K and 220 K; the dotted black mixing line is set to a threshold ambient temperature $T_A = T_C$, such that it makes tangential contact with $p_{liq}$ (diamond marker). The threshold ambient temperature defines the temperature above which contrail formation cannot take place. This is because mixing lines with $T_A > T_C$ will fail to contact $p_{liq}$, obviating CN activation. This condition forms the basis of the Schmidt-Appleman Criterion (Schmidt, 1941; Appleman, 1953).

In Fig. 1b, the ordinate has been transformed to the ice saturation ratio $S_i$ to clarify mixing behaviour when approaching the saturation lines. An additional dotted line has been added to this graph that corresponds to the "Koop line" (Koop et al., 2000), graphically this represents the boundary above which a system comprising aqueous droplets will freeze homogeneously. Using Fig. 1b, the freezing pathway for lubrication oil droplets will be explored. The point at which the lubrication oil droplets activate to form water droplets will be mapped onto the diagram and compared to the activation behaviour of nvPM, obtained from the literature. The activation behaviour of both lubrication oil droplets and nvPM will also



be related to the contrail mixing lines shown in Fig. 1. Experimentally observed freezing behaviour will be compared to the

Koop line and other relevant literature.

## 3. Materials and methods

This paper seeks to investigate the ability for ultrafine lubrication oil droplets to serve as ice-forming particles in contrails. This section outlines the techniques by which lubrication oil droplets were generated and characterised, and how their freezing behaviour was assessed.

### 3.1 Lubrication oil

The most abundant component of modern jet lubrication oils is a mixture of synthetic ester blends (Winder and Balouet, 2002; Craig and Barth, 1999). Additives are also present in jet lubrication oils, which improve their resistance to degradation (Duong et al., 2018), although these vary by manufacturer and specific application. AeroShell Turbine Oil 390 was selected for this study and contains a mixture of synthetic esters and additives. It has a low toxicity and a low viscosity (12.9 mm$^2$/s at 40°C),

which allows it to be atomized safely and effectively at low pressures.



## 3.2 Experimental setup

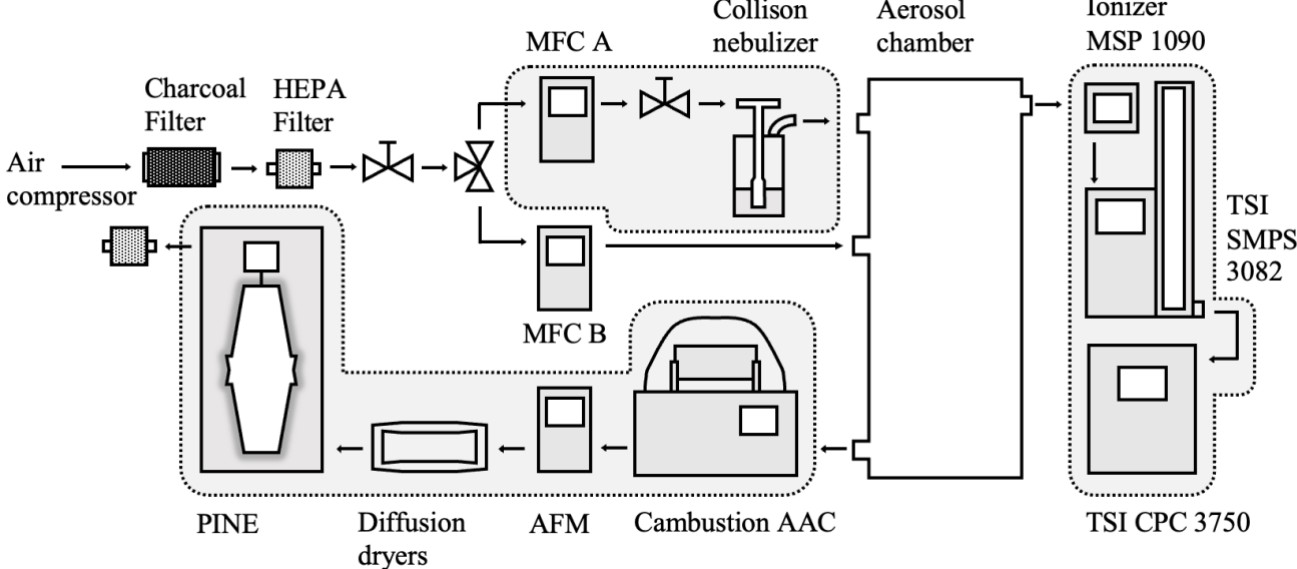


**Figure 2: The experimental setup. The flow of compressed air and/or aerosol around the system was driven by an air compressor and a series of vacuum pumps. Where possible, conductive tubing was used to reduce particle losses. An array of valves, mass flow controllers (MFC) and an aerosol flowmeter (AFM) were used to control the movement of air and/or aerosol through the system. Equipment is grouped into three dotted regions for clarity. The upper region highlights equipment used for generating lubrication**
**oil droplets, which were diluted using a 0.9 m³ aerosol chamber. The rightmost region highlights equipment used to perform size distribution measurements. This was enabled using a TSI ionizer (MSP 1090) and scanning mobility particle sizer (SMPS) (3938). The final (lower) region highlights equipment used to size-select the lubrication oil droplets and investigate their freezing mechanism(s). This was achieved using a Cambustion Aerosol Aerodynamic Classifier (AAC) and the prototype version of the portable ice nucleation experiment (PINE) expansion chamber (Möhler et al., 2021).**

The experimental setup used to perform measurements is illustrated in Fig. 2. This can be decomposed into five components: the inlet system, the droplet generation system (Sect. 3.3), the aerosol chamber, the size distribution measurement system and the expansion system (Sect. 3.4). The flow of aerosol through each of the five components was driven by a Titan Precision TT-25L-SILENT air compressor and a series of vacuum pumps. Before being exposed to sample aerosol, air was first cleaned by passing it through an activated carbon filter (Pall Carbon Capsule 12011) to remove residual organic material and then
through a Whatman high efficiency particulate air (HEPA) filter to remove ambient aerosol.

## 3.3 Droplet generation

Lubrication oil droplets were generated using a Collison 3-jet nebulizer (May, 1973). To minimise droplet concentrations, the nebulizer was run at its lowest operable pressure of 20 psig corresponding to a flow rate of approximately 5 L min$^{-1}$. Previous measurements with a similar jet lubrication oil indicated that under these conditions, particles were generated with an integrated
number concentration exceeding $10^6$ cm$^{-3}$. The condensation particle counter (CPC), TSI model 3750 and optical particle



counter (OPC), Palas Welas 2500 have integrated number concentration upper limits of $10^5$ cm$^{-3}$ and 4000 cm$^{-3}$ respectively. For this reason, several measures were taken to reduce the particle number concentration. Firstly, the Collison outlet was fed directly into a 0.9 m$^3$ aerosol chamber, whereupon aerosol was diluted with filtered air. Secondly, the compressed air inlet for the Collison was opened for a maximum duration of 1 s. These were denoted as "priming" events and were typically performed

every 100 min to maintain droplet number concentrations at least an order of magnitude greater than background number concentrations. For additional information, see Sect. S1 of the Supplement.

The flow of compressed air into the aerosol chamber was regulated at the inlet using a mass flow controller. To maintain the aerosol chamber at ambient pressure, the sum of flow rates in and out of the chamber were matched, except during "priming" events where the volume of aerosol passed into the chamber was negligible. The first aerosol chamber outlet was

connected directly to the particle-sizing apparatus, which comprised an electrical aerosol neutralizer (TSI model 1090) and Scanning Mobility Particle Sizer (SMPS, TSI model 3938) in series. The particle sizing apparatus allowed for the particle size distribution within the aerosol chamber to be measured continuously when interfaced with PINE.

### 3.4 Aerosol aerodynamic classifier and expansion measurements

The second aerosol chamber outlet was connected directly to the Cambustion aerosol aerodynamic classifier (AAC). The AAC

was used to select particles with a narrow range of aerodynamic diameters from the otherwise polydisperse particle size distribution in the aerosol chamber. The particle size distributions measured in the aerosol chamber and at the AAC outlet are related by the AAC transfer function, which describes the fraction of particles that successfully navigate the AAC as a function of particle diameter. Therefore, if both the AAC transfer function and the particle size distribution within the aerosol chamber are known, their product provides an indirect estimate for the particle size distribution at the AAC outlet (PINE inlet).

To characterise the transfer function, size distributions were recorded with and without the AAC present. During these separate measurements, an electrical mobility diameter ($d_m$) setpoint of 100 nm was selected using the AAC. To achieve this, the instrument was set to perform an internal numerical conversion between aerodynamic and mobility diameters, given the density and shape factor ($\rho$ = 924 kgm$^{-3}$ at 15 °C, $\chi$ = 1) of the oil droplets (Johnson et al., 2018). This resulted in an experimental transfer function with a geometric mean diameter (GMD) of $d_m$ = (100.9 ± 0.6) nm and geometric standard

deviation (GSD) of 1.08 ± 0.01, see Sect. S2 in the Supplement. The experimental transfer function was narrower than the theoretical AAC transfer function derived with inlet and size-dependent particle losses and resembled the idealised theoretical transfer function (Johnson et al., 2018). The narrowness of the experimental transfer function was attributed to the low vapour pressure, well-defined sphericity and density of the oil, which allowed for the conversion between $d_m$ and $d_a$ to be made with high confidence (Johnson et al., 2018).

During operation, the AAC outlet was connected to an inline aerosol flowmeter (AF10, Cambustion, UK). This was used to determine the flow rate upstream of the PINE system without interfering with the aerosol sample. Aerosol was then passed through a dual-membrane diffusion dryer to reduce its dew point and limit frost formation on the PINE expansion chamber walls (Möhler et al., 2021).



In terms of the expansion measurements, the PINE instrument comprises three main stages: inlet (diffusion dryers),
expansion chamber and OPC (Möhler et al., 2021). The expansion chamber has a capacity of 7 L and was cooled using a
Chiller Lauda (RP855) cooler. The gas temperature within the chamber was measured using three thermocouples which are
distributed vertically along the long axis, each of these were calibrated to a reference sensor with an accuracy of ± 0.1°K
(Möhler et al., 2021). The outlet flow passed through an OPC, which counted particles within a nominal diameter range of 0.6
- 40 μm. Each measurement using the PINE chamber comprised three modes: flush, expansion and refill (Möhler et al., 2021).
In the flush mode, approximately 21 L of aerosol flow was passed through the 7 L expansion chamber. This was to ensure that
the aerosol equilibrated with the chamber walls, so that immediately before an expansion, sample aerosol had a water partial
pressure equal to the vapour pressure above ice, $p_w = p_{ice}$ (100% $RH_i$). During the expansion mode, the inlet valve was closed
and the contents of the chamber were pumped out (via the OPC), resulting in adiabatic cooling and an increase in sample
relative humidity. Depending on the temperature of the chamber, water droplets and/or ice crystals were detected at the OPC
as a function of the temperature and pressure in the PINE chamber. Note that because the PINE chamber and OPC were
connected by a short (0.2 m) length of tubing, combining OPC (particle counts) and PINE (temperature/pressure) data based
on a common measurement time would introduce a non-negligible time delay due to particle travel. This time delay was
factored into the calculations of onset positions, resulting in a decrease in onset relative humidity values by 1%.

Measurements using the lubrication oil were performed using the experimental setup in Fig. 2. The aerosol chamber
was first "primed", generating average integrated chamber aerosol concentrations on the order of 1000 cm$^{-3}$. During these
measurements, the PINE chamber was cooled to initial chamber temperatures in the range 230 K < $T_i$ < 250 K. This allowed
for condensation and freezing events to be investigated in the temperature range 225 - 245 K, in line with the atmospherically-
relevant temperatures presented in Fig. 1. A continuous flow of aerosol was passed through the PINE system, with average
integrated aerosol concentrations on the order of 100 cm$^{-3}$ exiting the AAC.

**4. Results**

**4.1 Activation behaviour**

First, conditions required to activate ultrafine lubrication oil droplets to form water droplets were investigated. For each
expansion measurement, change-point analysis was used to identify the chamber pressure at which the OPC recorded a
significant change in particle count. Specifically, the point at which threshold values were met by both the particle
concentration and its time derivative was noted as the onset position. Combining the onset pressure with the pressure and
temperature before an expansion allowed for the onset temperature, $T_o$, to be determined (see Sect. S3 in the Supplement). The
$S_i$ at onset ($S_{i,o}$) could also be determined using these pressure and temperature values, enabling the phase space in Fig. 1 to be
populated with onset positions for each expansion. As mentioned in Sect. 2, the calculation of $S_i$ is sensitive to the
parametrization of $p_{liq}$. Two parametrisations were compared, and it was found that the more recent parametrization proposed
by Nachbar et al., (Nachbar et al., 2019) deviates from the parametrization proposed by Murphy and Koop (Murphy and Koop,



2005) by $< 1\%$ over the range of onset temperatures explored within this work ($T > 225.3$ K). The authors have therefore chosen to parametrize $p_{liq}$ according to Murphy and Koop.

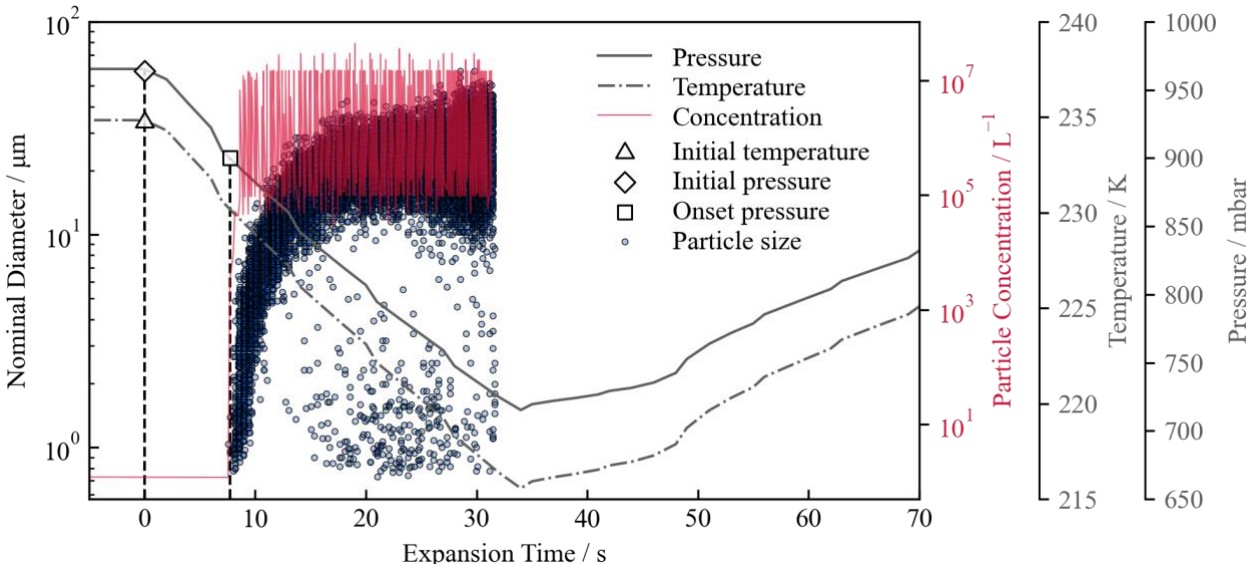

**Figure 3: Typical expansion recorded using the PINE chamber. The pressure and temperature of the PINE chamber have been**
**plotted against the duration into the expansion. Time delays discussed in Sect. 3.4 have been included. Additionally, initial and onset conditions have been noted using vertical dotted lines. The onset pressure was identified by applying change-point analysis to the particle concentration as described in the main text.**





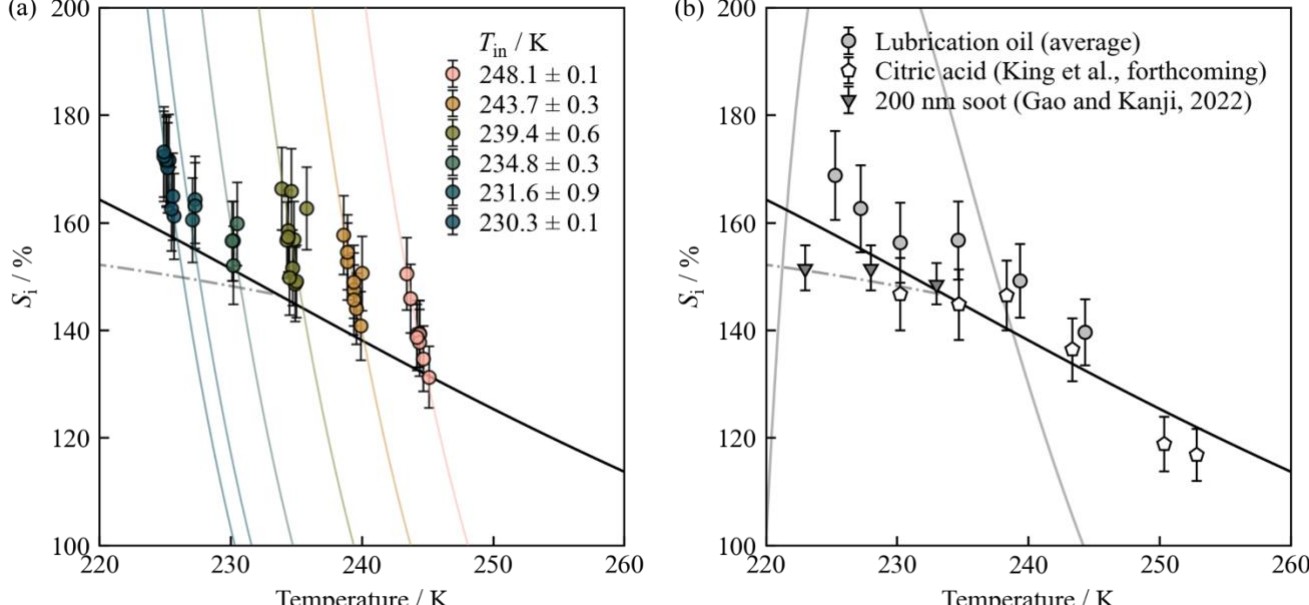

**Figure 4: (a) Expansion measurements performed using lubrication oil droplets with a GMD of $d_m = (100.9 \pm 0.6)$ nm. Each of the marker types represent a set of experimental measurements performed at similar initial chamber temperatures, with an average temperature $T_i$. Dry adiabats have been plotted for each of the marker sets, at the average initial temperature of the set, $T_i$. The onset temperature ($T_o$) and ice saturation ratio ($S_{i,o}$) were used to populate the phase space. (b) Experiments performed at similar initial temperatures were averaged (grey circles). For comparison, a contrail mixing line has been shown with $G = 1.64$ and $T_A = 220$ K which terminates at $p_{w,A} = p_{ice}$. Additionally, onset measurements for citric acid (King et al., forthcoming) obtained using the PINE chamber and 200 nm miniCAST soot (Gao and Kanji, 2022) obtained using the Horizontal Ice Nucleation Chamber (HINC) have been marked on the phase space. Uncertainties associated with onset temperature measurements are discussed in the Supplement, Sect. S3. Accessible colourmap obtained from (Crameri, 2018).**

Each of the marker types in Fig. 4a correspond to a set of repeat measurements with the PINE chamber at a similar initial temperature (with an average temperature of the set $T_i$). Uncertainties in the initial pressure values were estimated using the standard deviation of the final 300 equilibrated pressure values recorded prior to an expansion. Uncertainties in the onset pressure values were limited by the instrumental resolution (1 Hz) and determined conservatively as half the average pressure difference between adjacent values during the expansion. Pressure and temperature ($\pm 0.1$ K) uncertainties were used to derive the error bars on the onset saturation ratio ($\Delta S_{i,o}$). For a given $T_i$, these pressure uncertainties lead to onset positions being distributed along the corresponding dry adiabat, accounting for the trend shown in Fig. 4a.

Repeat experiments performed with similar initial chamber (and therefore onset) temperatures were averaged to produce Fig. 4b. For comparison, onset positions for citric acid aerosol have been shown on Fig. 4b (King et al., forthcoming). These were obtained using the PINE chamber and the experimental data were analysed according to the relations outlined in Sect. S3 of the Supplement. The onset positions for citric acid aerosol follow the expected behaviour for a hygroscopic aerosol (Koop et al., 2000): they are positioned along the water saturation line until the bifurcation at ~234 K, after which they extend




along the Koop line. This agreement affords confidence in the onset positions for lubrication oil droplets obtained in this work, since these were investigated using similar methods.

The onset positions of lubrication oil droplets lie above the water saturation line for all temperature investigated (225 – 245 K). These onset positions are consistent with the limiting behaviour of a hydrophobic, insoluble aerosol and are characterized by a hygroscopicity parameter $\kappa$ that approaches 0, in accordance with $\kappa$-Köhler theory (Petters and Kreidenweis, 2007), see Sect. S4 of the Supplement. This result was anticipated on account of the lubrication oil's negligible water solubility (Sullivan et al., 2009). To place the behaviour of lubrication oil in the context of contrail formation, a comparative study on the onset behaviour of aircraft-generated nvPM is required.

Despite the wealth of literature concerning the onset behaviour of nvPM (Mahrt et al., 2018; Koehler et al., 2009; Möhler et al., 2005; Gao et al., 2022b), there is a paucity of information concerning the onset behaviour of aircraft-generated nvPM, or representative surrogates (Marcolli et al., 2021). The example nvPM onset positions presented in Fig. 4b. are taken from recent experiments performed using size-selected miniCAST-generated soot with a modal diameter of 200 nm (Gao and Kanji, 2022a). This study sits within a body of research that provides evidence in support of the pore condensation and freezing (PCF) mechanism for large (100 - 400 nm) porous soot aggregates (Marcolli et al., 2021; David et al., 2019). During PCF, water is collected within mesopores (2 - 50 nm) by capillary condensation under *subsaturated* conditions, after which it freezes. To our knowledge, PCF has not been measured for soot particles with sizes that are representative of aviation nvPM, and it is challenging to extrapolate the implications of PCF to representatively-sized nvPM, particularly given additional complications associated with onset sensitivity to internal and external mixing of semi-volatile material (Gao and Kanji, 2022; Gao et al., 2022b, a). Given the onset positions for lubrication oil droplets follow the extremal curve for an insoluble aerosol, and based on the available literature referenced above, it is reasonable to conclude that lubrication oil droplets will activate at supersaturations no lower than those required to activate aircraft-generated nvPM. In other words, aircraft-generated nvPM particles are likely to be more effective CN than ultrafine lubrication oil droplets.

During contrail mixing, modest deviations from $T_C$ result in peak supersaturations of ~10% (Bier et al., 2022; Kärcher et al., 2015), depending on the properties of nascent aerosol in the plume. According to Fig. 4b., these conditions would be sufficient to activate a proportion of both nvPM particles and lubrication oil droplets. For contrails forming below threshold conditions ($T_A \ll T_C$), it is therefore feasible that lubrication oil droplets could competitively deplete plume supersaturations during contrail mixing under soot-poor conditions. However, under near-threshold conditions ($T_A \sim T_C$), where only infinitesimal supersaturations are achieved, it is likely that nvPM will preferentially activate in favour of the less active lubrication oil droplets. These implications have previously been corroborated computationally (Kärcher and Yu, 2009).





## 4.2 Freezing behaviour

As discussed in Sect. 4.1, the activation of lubrication oil droplets is important for predicting their behaviour under contrail-forming conditions. Additionally, the freezing mechanism(s) for these droplets, and the temperatures at which these mechanism(s) are operable, are also important for understanding the behaviour of resulting ice crystals under contrail-forming conditions.



**Figure 5: Nominal particle size distributions as a function of temperature for representative expansions at the onset temperatures denoted. Accessible colourmap obtained from (Crameri, 2018).**

To elucidate the freezing mechanism(s), particle size distributions were investigated as a function of temperature through individual expansions, these are shown in Sect. S5 of the Supplement. Four distributions have been displayed in Fig. 5, which serve to disambiguate the freezing mechanism(s). The expansion corresponding to $T_o = 244.2$ K produced a monomodal distribution with a GMD of $(5.25 \pm 0.01)$ µm and GSD of $1.100 \pm 0.002$, resulting from the activation of lubrication oil droplets into water droplets (as described in Sect. 4.1). For the expansion with an onset temperature $T_o = 240.0$ K, the



droplet mode was again visible at a comparable GMD of $(3.4 \pm 0.1)$ μm (GSD = $1.72 \pm 0.05$), persisting down to a temperature of ~235 K. However, when the temperature within the chamber decreased below this value, the droplet mode disappeared, and a second particle mode appeared at a larger GMD of $(15.4 \pm 0.1)$ μm (GSD = $1.47 \pm 0.01$). Owing to the relative size difference

between the two modes (Lacher et al., 2017), the latter mode can be attributed to ice crystals which form when a fraction of the activated lubrication oil droplets subsequently freeze.

During the next coolest expansion ($T_o$ = 230.2 K), the particle size distribution grew to a GMD of approximately $(21.6 \pm 0.1)$ μm (GSD = $1.276 \pm 0.005$). The expansion at $T_o$ = 224.9 K demonstrates similar behaviour with a comparable distribution width GSD = $(1.27 \pm 0.01)$, albeit with a smaller GMD of approximately $(14.5 \pm 0.1)$ μm. The size distributions for the two

expansions at cooler temperatures feature the familiar ice crystal mode at larger nominal particle diameters (> 5 μm). Based on the analysis presented in Sect. 4.1, condensation must have taken place shortly after water saturation was reached. This implies that during the two expansions performed at cooler temperatures, ice crystals nucleated from growing water droplets. This process is referred to as condensation and freezing and has been well-documented in the literature (Murray et al., 2012)

Based on the evidence in Fig. 5, the microphysical pathway to ice crystal formation has been partly justified as

condensation followed by freezing. Supercooled water droplets containing aerosol particles can freeze either heterogeneously or homogeneously. Homogeneous freezing occurs when water molecules on the surface of the aerosol particle reach a critical cluster size, which depends upon the droplet volume, temperature and freezing rate. Once the critical cluster size is reached, the rate of homogeneous freezing becomes comparable to the experimental timescale and freezing occurs readily (Murray et al., 2012). The homogeneous freezing temperature for micrometre-sized droplets has been demonstrated to lie in the range

(235 - 236) K, as verified by numerous experimental studies using a variety of techniques (Tanaka and Kimura, 2019; Zhang et al., 2012; Murray et al., 2010; Tarn et al., 2021). Using the methods outlined in (Koop and Murray, 2016; Murray et al., 2010), the onset of homogeneous freezing under conditions in Fig. 5b was estimated as ~235 K, see Sect. S6 of the Supplement. In contrast, for a water droplet to freeze heterogeneously, the immersed aerosol particle must have surface properties that lower the activation energy to ice nucleation (Knopf et al., 2018; Knopf and Alpert, 2023). For this reason, heterogeneous freezing

temperatures are necessarily warmer than homogeneous freezing temperatures, under otherwise comparable conditions (Kärcher and Lohmann, 2003).

It is important to understand the freezing mechanism adopted by contrail ice-forming particles as this influences the size distribution of ice crystals and hence the optical properties of the contrail. To demonstrate this, consider a parcel of air cooling in the atmosphere, consisting of both heterogeneous and homogeneous ice-forming particles. If we assume that both particle

types activate to form water droplets under similar conditions, cooling droplets that contain the heterogenous ice nuclei will freeze preferentially, as reasoned above. Computational and experimental studies on cirrus cloud formation suggest that because these freezing events deplete the supply of water vapour, they can act to delay or even inhibit homogeneous freezing in the remaining droplets (DeMott et al., 2003; Spichtinger and Gierens, 2009). As there is less competition for growth, the ice crystals will be larger than those that would have formed had all the droplets frozen homogeneously; however, because only a

fraction of the droplets have frozen, there will be fewer ice crystals overall (Spichtinger and Gierens, 2009). As a consequence,





the total droplet surface area is reduced, which is an important parameter controlling optical depth, and thus radiative forcing (Frömming et al., 2011; Kärcher, 2016). Note that these conclusions were drawn from experimental and computational measurements concerning cirrus clouds, which have rates of $S_i$ and concentrations of ice-forming nuclei that differ by several orders of magnitude from those experienced within a contrail mixing plume. The above discussion therefore serves only to
illustrate that the freezing mechanism adopted by ice-forming particles within a cooling air parcel influences its optical properties.

In Fig. 5b, freezing is observed during water droplet growth at a temperature of ~235 K, which lies within the expected range for homogeneous freezing and is consistent with the theoretical value identified using Fig. S5 in the Supplement. Therefore, it is likely that water condensed on the lubrication oil droplets freezes homogeneously rather than heterogeneously.
Finally, the microphysical pathway to ice crystal formation on lubrication oil droplets has been justified as condensation followed by homogeneous freezing. This pathway is illustrated pictorially in Fig. 6.

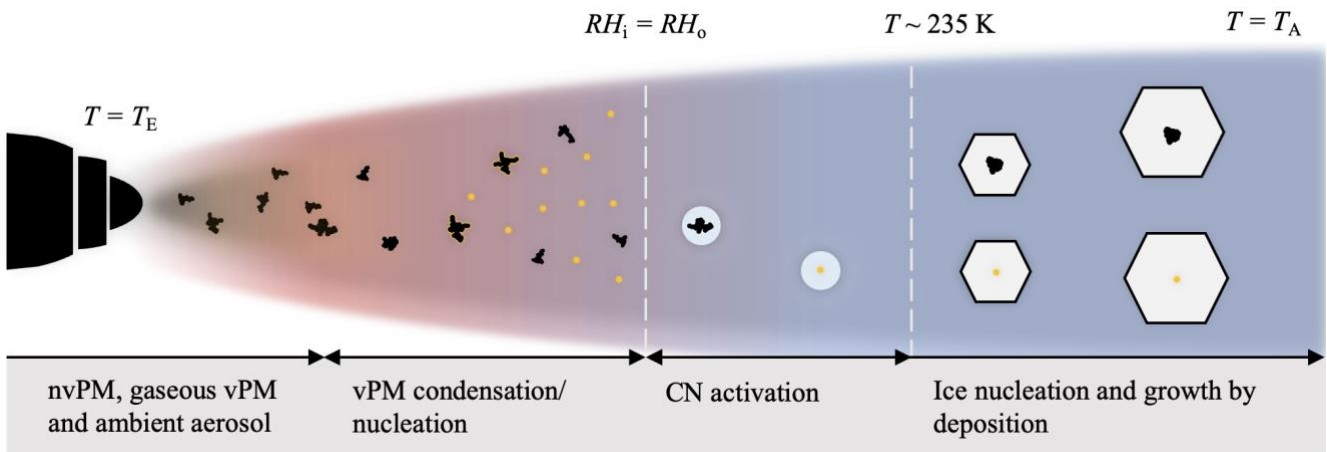

**Figure 6: Freezing pathway adopted by lubrication oil droplets (condensed vPM, yellow) and nvPM particles during contrail mixing. Key microphysical (phase) transitions are represented pictorially with increasing plume dilution between temperatures $T_E$ and $T_A$.**
**Blue circles represent activated soot/lubrication oil droplets. These later freeze to form ice crystals, which grow by deposition of supersaturated water vapour. Dimensions are not to scale.**

## 5. Conclusions

In this paper, the freezing pathway for ultrafine jet lubrication oil droplets was experimentally determined under contrail-forming conditions, using the PINE expansion chamber (Möhler et al., 2021). The principal objective was to examine whether
lubrication oil droplets could contribute towards contrail formation in the soot-poor regime or indeed in the absence of nvPM emissions (i.e., aircraft powered by hydrogen and electric aircraft).

Lubrication oil droplets were generated using a Collison nebulizer and size-selected using an AAC, producing a lognormal particle size distribution with a GMD of $d_m = (100.9 \pm 0.6)$ nm and GSD of $1.17 \pm 0.01$. Measurements were





performed in the temperature range 225 - 245 K and lubrication oil droplets were found to exhibit water activation under
conditions consistent with that of a hydrophobic, insoluble aerosol (Petters and Kreidenweis, 2007). The hygroscopicity
parameter is an intrinsic material property, therefore the activation behaviour exhibited by ultrafine lubrication oil droplets is
adaptable to larger droplets (Yu et al., 2010) or fractional oil coatings on nvPM (Yu et al., 2012) using $\kappa$-Köhler theory (Petters
and Kreidenweis, 2007), for use in computational models. Following activation and growth, droplets were shown to freeze
homogeneously at ~235 K, in accordance with the homogeneous freezing temperature for micrometre-sized water droplets
(Murray et al., 2010), see Sect. S5 of the Supplement. Therefore, the microphysical pathway to ice formation for lubrication
oil droplets was shown to be condensation followed by homogeneous freezing (Murray et al., 2012).

Threshold plume temperatures (see Sect. 2) are typically cooler than the homogeneous freezing temperature, therefore
it follows that condensation followed by homogeneous freezing is also thought to be the dominant mechanism for the formation
of contrail ice crystals from aircraft-generated nvPM (Kärcher et al., 2015). This microphysical pathway to ice formation
physically underpins the Schmidt-Appleman Criterion. Although lubrication oil droplets were shown to be less effective CN
than nvPM presented in the literature (Gao and Kanji, 2022), the critical supersaturations required to activate both species are
readily achievable for below-threshold conditions (see concluding remarks in Sect. 4.1). Therefore, for below-threshold
conditions and with reduced nvPM particle emissions, ultrafine lubrication oil droplets could feasibly compete with nvPM for
plume supersaturations, forming contrail ice crystals. Critically, to account for the behaviour of hydrophobic aerosol species
that activate above water saturation and for hygroscopic species that activate below water saturation (e.g., via PCF), the
Schmidt-Appleman Criterion may need to be revised to accurately describe contrail formation.

In addition to their central role in contrail formation, soot particles emitted by aircraft are also thought to affect
background cirrus properties by providing sites on which nucleation of ice can occur (Kärcher et al., 2021; Mahrt et al., 2020).
To that end, it is reasonable to question whether lubrication oil droplets would be capable of modifying the properties of
background cirrus. The results of this study demonstrate that lubrication oil droplets only serve as ice-forming particles above
water saturation, hence they will not compete with background hygroscopic aerosol or natural ice-nucleation particles such as
mineral dust or black carbon (Cziczo et al., 2013), which can produce ice crystals well-below water saturation in concentrations
sufficient to limit the supersaturation. Therefore, unless lubrication oil droplets become appreciably more hygroscopic through
atmospheric (photo)chemical aging processes, they will serve only to enhance contrail ice crystal numbers and will not interfere
with background cirrus formation.

The potential for lubrication oil droplets to serve as an effective source of contrail ice-forming particles also depends
upon their relative emission index and size distribution at cruise altitude. Accurate estimates for the size distribution and
emission indices of both lubrication oil droplets and ambient aerosol would enable effective ice crystal emission indices to be
deduced in the soot-poor regime, which would benefit from the results obtained within this paper (Bier and Burkhardt, 2019;
Schumann, 2012). Global simulations demonstrate that a two-fold reduction in initial contrail ice crystal numbers would lead
to a decrease in radiative forcing by 20% (Burkhardt et al., 2018). Therefore, it is paramount that the emission index of
lubrication oil during cruise conditions is characterised. Nevertheless, efforts to reduce lubrication oil number emissions in the

exhaust plumes of jet engines would help to reduce the climate impacts of contrails, thereby reducing the climate impact of aviation.

## Competing Interests

The authors declare that they have no conflict of interest.

## Author Contribution

All authors conceptualized the project. LK provided training for use of the PINE chamber and original code for the analysis of data. JP designed the experiments, conducted the measurements, and collected the raw data. JP wrote the original manuscript. All authors reviewed and edited the manuscript. MEJS and BJM supervised the project.

## Acknowledgements

Instrumentational support was funded through the Natural Environment Research Council (NERC) Atmospheric Measurement and Observational Facility (AMOF), grant AMOF_20220527111415. The PINE chamber was developed with support from the European Research Council (Marine Ice grant number. 648661). The AAC was funded by the UK Engineering and Physical Sciences and Research Council (EPSRC, EP/T024712/1). JP was supported by the EPSRC CDT in Aerosol Science (EP/S023593/1). LK was supported by the NERC PANORAMA DTP (NE/S007458/1).

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
