# Peer review of "Jet aircraft lubrication oil droplets as contrail ice-forming particles"

_EGUsphere, 2023_

## Author Comment (AC1)

**Response to Reviewer Comments**

We thank the reviewers for their comments, which have helped us to improve the quality of the manuscript. In this document, reviewer comments are indicated using italic text. Our responses are written using normal (non-italic) text. When page and line numbers are given, these refer to the revised manuscript unless otherwise stated. Normal blue text is used to indicate text cited from the revised manuscript and **bold blue text** is used to indicate new material introduced into the manuscript.

**Referee 1 (RC1)**

1. *In this paper, the authors investigated the homogeneous freezing properties of lubrication oil droplets using a PINE expansion chamber. These laboratory experiments conclude that oil droplets freeze at temperatures below 235K and suggest that reducing the number concentration of these droplets could mitigate the climate impact of aircraft contrail cirrus clouds. Detailed understanding of contrail ice clouds is currently poorly understood and also poorly represented in the model. This study aims to fill this gap; however, it lacks originality. The main concerns are as below.*

   *Page 93-107: There are no direct measurements that support that lubrication oil droplets are found in the exhaust plumes. In fact, as per Kärcher et al. (2016) lubricant-derived aerosol particles are too few and do not influence contrail formation. Measurement conditions (ground level) are significantly different from high-altitude aircraft cruising altitudes.*

We acknowledge that emissions produced at ground level may differ from those produced at cruise altitude. For this reason, we express the need for characterization of lubrication oil emissions at cruise altitude twice in the original manuscript. These instances can be found at: lines 412 – 413 (main text) of our concluding paragraph where we stated that:

"it is paramount that the emission index of lubrication oil during cruise conditions is characterized",

and (b) line 18 (main text) in our abstract where we stated that:

"[computational] studies would benefit from particle size distribution measurements at cruise altitude".

To further emphasise this point, we have introduced the following statement to the introduction, immediately before our literature review on lubrication oil emissions [main text: lines 84 – 87]:

**The following section outlines current scientific understanding of lubrication oil as a component of aircraft emissions. The observations presented below have been obtained from ground-based measurements, which may not directly translate to cruise conditions. Nevertheless, lubrication oil droplets are expected to be produced during cruise, but their production rates remain uncertain and are likely to vary significantly across engine types.**

The paper cited by the reviewer (Kärcher, 2016) states that "aircraft jet engines may also emit metal particles and lubricant aerosol particles depending on maintenance and power setting, but those particles are too few by number and occur too intermittently to explain contrail formation". This paper is referring to soot rich engine emissions, whereas our paper is very clearly about what happens in the absence of soot. Indeed, a contribution of our paper is to

demonstrate that under soot-rich conditions, lubrication oil droplets are unlikely to compete with soot for plume supersaturations. However, under soot-poor conditions, where the relative number concentration of lubrication oil droplets is increased (with respect to the number concentration of soot particles) our results suggest that lubrication oil droplets may compete with soot particles, depending on their size distribution at cruise altitude (see main text: lines 301 – 307). Therefore, we provide original evidence in support of the soot-poor hypotheses proposed by Kärcher in a later publication (Kärcher, 2018), thus making a significant contribution to understanding the role of other aerosols on contrail formation, particularly under soot-poor conditions.

Given no one has experimentally examined if jet lubrication oils can activate to ice in contrails, working with nebulised lubrication oil - as done in our work - is a robust and necessary first step. We highlight the originality of our research twice in the manuscript at the following points: (a) lines 11 – 13 (main text):

"Ultrafine (<100 nm) jet lubrication oil droplets constitute a significant fraction of the total organic particulate matter released by aircraft, however their ability to form contrail ice crystals is hitherto unexplored."

and (b) lines 107 – 108 (main text):

"the impact of lubrication oil droplets on contrails is hitherto unexplored, this paper addresses their ability to function as contrail ice-forming particles."

> 2. *The atmospheric relevance of lubrication oil (section 3.1) used as a surrogate for the actual oil (if any) is not clear. Is the oil used (section 3.1) found in the actual exhaust plume? Chemical composition of oil droplets sampled downstream of the exhaust plume and surrogate oil droplets used in this study should be compared. Such experiments are missing in this paper.*

Aircraft lubrication oils are identifiable with a unique chemical signature using mass spectroscopy (Yu et al., 2012), as they derive from a similar base stock of synthetic esters. This is discussed in the original manuscript [lines 79 – 81 (main text)]:

"in the literature, the contribution of jet lubrication oil derivatives towards the total organic exhaust fraction is commonly quantified using the ratio of ion fragment intensity at m/z = 85 and 71, obtained using mass spectrometry (Yu et al., 2012)."

For this reason (as discussed in Sect. 3.1) our experiments were performed using a commercially available aero engine lubrication oil composed of a synthetic ester blend.

Previous studies have demonstrated that the chemical composition of aerosols sampled in the plume of an aircraft can be related to the lubrication oil used by the engine (Yu et al., 2010; Timko et al., 2010; Yu et al., 2012). These studies confirm that lubrication oil is present in the exhaust plume of aircraft engines. We first made this point in the original manuscript in lines 87 - 94, and have updated this text to include the measured range of emissions indices [main text: lines 87 - 94]:

"Lubrication oil can be released from overboard breather vents (Eastwick et al., 2006; Nie et al., 2018) or via clearance seals (Flitney, 2014), which form part of the aircraft oil recirculation system (Hunecke, 2003). To that end,  **the emission index of oil sampled directly from breather vents at engine idle has been reported in the range 2 - 12 mg kg$^{-1}$ (Yu et al., 2010). Additionally,** measurements

performed by another group, 15 m downstream of an engine exit plane, found that a significant proportion of lubrication oil existed in the particle size range > 300 nm (Timko et al., 2010). The researchers found that for a different engine, 90% of the condensed vPM mass derived from lubrication oil and was confined to a particle size range 80 – 500 nm, qualifying that the characteristics of lubrication oil emissions are sensitive to engine technology."

In the manuscript, we corroborate these findings by referring a study by Yu et al., (Yu et al., 2012) that employed a mobile laboratory to sample downwind of an active runway. The results of this study are summarised in the original manuscript where we write that [lines 94 – 96 (main text)]:

"measurements taken 30 – 150 m from active taxiways also identified lubrication oil contributions towards vPM between 5% and 100% in the particle size range 50 – 700 nm, in association with the nvPM particle mode (Yu et al., 2012)."

In the cited study, emission measurements were recorded during individual "plume events" that correspond to distinct aircraft. Across a total of 12 plume events, the authors report a median emission index of lubrication oil of 8.3 mg kg$^{-1}$ (range of 0.4 ± 0.5 mg kg$^{-1}$ – 255 ± 50 mg kg$^{-1}$) corroborating the previous study (Yu et al., 2010) and indicating that the mass-based emissions of lubrication oil could be substantial for some engines.

In addition, the particle size distribution recorded during the most organic-rich plume event has a geometric mean diameter of approximately 10 nm, which is consistent with several other studies that report significant lubrication oil contributions in the nucleation mode (Yu et al., 2019; Ungeheuer et al., 2020; Fushimi et al., 2019). These were discussed at length in the manuscript [lines 98 – 103 (main text)]:

"Measurements taken 30 m downstream (of the engine exit plane) of an aircraft operating at 85% thrust have demonstrated that lubrication oil is the dominant contributor towards vPM, particularly in the nucleation mode (< 30 nm) (Yu et al., 2019). This was corroborated by near-runway sampling at Narita and Frankfurt International Airports (Fushimi et al., 2019; Ungeheuer et al., 2020), where researchers found that the majority of compounds detected in nucleation mode particles (respectively defined as < 30 nm, 10 – 56 nm) could be attributed to jet lubrication oil components."

3. *Section 3.1, 3.2: The experimental setup to generate the oil droplets is not similar as the actual aircraft engine generating the oil droplets. Does the aerosol generation mechanism change the chemical properties of oil droplets? The physical and chemical properties of generated oil droplets within this paper are similar to actual exhaust plume? Note that the aircraft engine is operated at different thermal and turbulent conditions. The study will be unique if actual aircraft engine exhaust plume is sampled.*

The mechanisms for production of lubrication oil droplets in aircraft engines are not yet well-understood, at least in the academic literature. The engine lubrication system and oil used differs depending on the engine model and manufacturer. For this reason, it is challenging to identify a standard lubrication oil size distribution, see (Timko et al., 2010).

There may be several production pathways that range from venting of oil mist directly to atmosphere via a vent on the nacelle of the engine or venting of oil mist into the turbine, where it may undergo thermally-driven processes such as degradation, evaporation and condensation, which may change the oil's chemical properties (Wang et al., 2004). We highlight these points in the original manuscript in lines 93 - 94 (main text), where we reference measurements performed by Timko et al., (Timko et al., 2010), stating that [lines 93 - 94 (main text)]:

"the characteristics of lubrication oil emissions are sensitive to engine technology".

As discussed, the extent of any chemical changes will depend on the specific engine lubrication oil system. To illustrate this, researchers sampling oil emissions directly from breather vents found that mass spectra obtained from these measurements were comparable to those obtained from nebulized oil (Yu et al., 2010). Overall, we hope that our study inspires future studies using aerosol that has been exposed to varying engine conditions and also perhaps actual engine emissions from a test engine, or an aircraft.

Although we size-selected the lubrication oil droplets in order to understand the activation and freezing behaviour, both these properties can be parametrized for other particle sizes via the Kelvin equation (Petters and Kreidenweis, 2007) and relations for homogeneous freezing provided in Sect. S7 of the Supplement. (Murray et al., 2010b). We discuss this in the original manuscript [lines 379 - 382 (main text)]:

"The hygroscopicity parameter is an intrinsic material property, therefore the activation behaviour exhibited by ultrafine lubrication oil droplets is adaptable to larger droplets (Yu et al., 2010) or fractional oil coatings on nvPM (Yu et al., 2012) using $\kappa$-Köhler theory (Petters and Kreidenweis, 2007), for use in computational models."

We intend to repeat these measurements using heated (recondensed) lubrication oil, and also lubrication oil in combination with other volatile/semi-volatile particulate matter such as sulfuric acid (Ungeheuer et al., 2022). This will enable us to examine the activation and freezing behaviour of more complex volatile/semi-volatile emissions. To that end, we have introduced the following sentences into the conclusion [main text: lines 402 - 406]:

"Therefore, unless lubrication oil droplets become appreciably more hygroscopic through atmospheric (photo)chemical aging processes, they will serve only to enhance contrail ice crystal numbers and will **likely** not interfere with background cirrus formation. **We intend to repeat these measurements using different droplet generation pathways and in combination with other aircraft plume aerosol, to examine the activation and freezing behaviour of more complex emissions.**"

4. *Section 4.1: The setup shown in Figure 2 describes that diffusion dryers are used upstream of the PINE chamber. If the air is dried, then what is the source of humidity in the PINE chamber? If correct, there must be some source of water vapor to activate oil into water droplets. What is the RH of the air upstream (after dryers) and within the PINE chamber? If the air is not completely dried (maybe RH = 5% then the dewpoint temperature is -21 degC of 20 degC room temperature air that is entering the chamber), then as PINE is cooled most of the incoming water vapor will condense on the interior walls of the PINE chamber as soon dew point is reached instead on the oil droplets. This will result in very few droplets activation (Figure 3). What is the droplet activation fraction at saturated supercooled temperature conditions? Such thermodynamic analysis (trajectory analysis) of oil droplets (particles) that are sampled is missing. As there are no RH measurements within the PINE, it is difficult to understand the activation behavior of droplets. Also, this poor understanding makes other ice nucleation groups to reproduce these results.*

The reviewer has correctly stated that the diffusion dryers do not remove all of the water vapour before the PINE aerosol inlet. Indeed, prior to experimental work the driers were calibrated such that there would be sufficient water vapour to reach (and go beyond) water-saturated conditions during the expansions. Importantly, the relative humidity of the inlet aerosol must be sufficiently high to enable droplet activation during expansion, but low enough to limit frost formation on the chamber walls, see Sect. 3 in (Möhler et al., 2021).

As discussed in the Supplement Sect. S3, this means that at the start of the expansion, aerosol within the chamber has a partial pressure of water equal to the partial pressure of ice. Therefore, although the relative humidity within the chamber is not measured directly, it can be calculated at any point (pressure, temperature) during the expansion using this starting condition and the adiabatic approximation. The fact that droplet onset for citric acid (a soluble hygroscopic material) occurs around water saturation (Fig. 4b), shows that these assumptions are valid and that we have a good understanding of the relative humidity in the chamber.

5. *Section 4.2: The freezing behavior of oil droplets has been widely studied in the past. See Tabazadeh et al. (2002) and many papers that cite this work. It is very well known the T and RH conditions where these oil droplets freeze homogeneously. The results shown in Figure 4 are well described in the literature. It is not clear the uniqueness of the lubricant oil that is used in this study. As mentioned above, the atmospheric relevance is missing.*

The literature cited by the reviewer (Tabazadeh et al., 2002) investigates ice nucleation in water droplets that are suspended in oil. In our work we investigate the condensation of water onto aerosolised oil droplets followed by freezing. Our experiments enable us to investigate the point at which we observe activation of lubrication oil droplets within a supersaturated atmosphere, and the conditions under which ice nucleation takes place. The experiments of Tabazadeh et al. (2002) offer no insight to the ice forming potential of lubrication oil droplets under contrail conditions. This objection seems to be the origin of the referee's comment that our study 'lacks originality'. This comment seems to have been made on the basis of a misunderstanding.

**Referee 2 (RC2)**

1. *The manuscript titled "Jet aircraft lubrication oil droplets as contrail ice-forming particles" by Ponsonby et al., investigates water and ice nucleation affected by lubrication oil droplets relevant to aircraft engines. Basic thermodynamics of supersaturation conditions are first presented followed by nucleation results. The authors claim cloud condensation nuclei activation with a hygroscopic parameter close to 0, and claim that ice nucleation occurred homogeneously. The contrail mixing line is cleverly plotted together with their results to highlight the importance of T and RH ranges at which ice or liquid could nucleate when exhaust plumes mix with colder dryer ambient air.*

   *The study is performed well, and the conclusions are sound. The negative result that particles nucleate with no hygroscopicity and homogeneously is important to be published. It will guide future work on used oil and help to aid in interpreting ice nucleation from aircraft emitted particles. Certainly, it also shows the suitability of the PINE instrument to measure nucleation in general. The paper is written and presented well, and suitable for publication. I only have a few minor comments to be addressed.*

   *Figure 1: It is not so clear what is water partial pressure and saturation vapor pressure. Would the authors please identify this? pw,x is the saturation vapor pressures where x is i for ice or w for water. The lines are predicted water vapor partial pressure. It would be appreciated for this to be claimed in the caption.*

We thank the reviewer for noticing this inconsistency. We have changed the label on the y-axis to represent the partial pressure of water vapour ($p_w$) and have ensured that the notation used for saturation vapor pressures is consistent in the text and in the legend. Below is the location where changes have been made:

[main text: Fig. 1a]:

[Figure]

2. *Figure 1 and p5 l126-127: G is the slope of the colored lines in Fig. 1A. Although claimed that G=1.64 is a typical value, it would be beneficial to show or state a range. I would expect that the different types, manufactures and sizes of engines, G may have different values.*

Thank you for your suggestion. You are correct in stating that the value of $G$ is sensitive to the aircraft and fuel type. For aircraft burning Jet-A1 kerosene, variations in the slope parameter $G$ (see Eq. 1) result principally from differences in the overall propulsion efficiency ($\eta$) between different aircraft. This parameter is typically estimated as 0.3 with a range of (0.2, 0.4) (Teoh et al., 2022). We agree that it would be helpful to illustrate the typical range of values of $\eta$ exhibited by different aircraft burning conventional fuel and have introduced the following sentences [main text: lines 130 – 135]:

"Each contrail mixing line is described by $G = 1.64$, corresponding to typical aircraft and fuel properties ($p_T$, $EI_w$, $Q$, $\eta$) of (250 mb, 1.23 kg.kg$^{-1}$, 43.2 MJ, 0.3) (Kärcher et al., 2015) and terminates at $p_{w,A} = p_{ice}$. **Note that for aircraft burning kerosene-based fuels, $\eta$ can range between 0.2 and 0.4 (Teoh et al., 2022) depending on aircraft and engine type. This results in an indicative range of $G$ between 1.10 and 2.19, which can impact contrail formation** (Schumann, 2000). For the  **coloured** mixing lines, $T_A$ takes values of 210 K, 215 K and 220 K; the dotted black mixing line is set to a threshold ambient temperature $T_A = T_C$, such that it makes tangential contact with $p_{liq}$ (diamond marker)."

3. *Figure 1 and p8 l196: Would it be worth to state or show where homogeneous liquid nucleation (kelvin equation) would occur? Supersaturation with respect to water is predicted to be very high, and it would be interesting to show where nucleation on 100nm particles that are completely hygroscopic would occur.*

Thank you for your suggestion. We agree that it would be useful to show the reader where heterogeneous nucleation of water on a perfectly wettable (albeit non-hygroscopic) material would occur. We have introduced this in Fig. 4b and introduced the following sentences [main text: Fig. 4b]:

[Figure]

[main text: Fig. 4b caption]:

"Figure 4: (a) Expansion measurements performed using lubrication oil droplets with a GMD of $d_m$ = (100.9 ± 0.6) nm. Each of the marker types represent a set of experimental measurements performed at similar initial chamber temperatures, with an average temperature $T_i$. Dry adiabats have been plotted for each of the marker sets, at the average initial temperature of the set, $T_i$. The onset temperature ($T_o$) and ice saturation ratio ($S_{i,o}$) were used to populate the phase space. (b) Experiments performed at similar initial temperatures were averaged (grey circles). For comparison, a contrail mixing line has been shown with $G$ = 1.64 and $T_A$ = 220 K which terminates at $p_{w,A} = p_{ice}$. Additionally, onset measurements for citric acid (King et al., forthcoming) obtained using the PINE chamber and 200 nm miniCAST soot (Gao and Kanji, 2022) obtained using the Horizontal Ice Nucleation Chamber (HINC) have been marked on the phase space. **The Kelvin line has also been plotted, representing a graphical boundary above which wettable, insoluble particles will activate**. Uncertainties associated with onset temperature measurements are discussed in the Supplement, Sect. S3. Accessible colourmap obtained from (Crameri, 2018)."

[main text: lines 281 – 286]:

"The onset positions of lubrication oil droplets lie above the water saturation line for all temperature investigated (225 – 245 K). These onset positions are consistent with the  **Kelvin equation, which represents the limiting activation behaviour of wettable, insoluble aerosol particles, which** are characterized by a hygroscopicity parameter $\kappa$  **= 0**, in accordance with $\kappa$-Köhler theory (Petters and Kreidenweis, 2007), see Sect. S4 of the Supplement. This result was anticipated on account of the lubrication oil's negligible water solubility (Sullivan et al., 2009). To place the behaviour of lubrication oil in the context of contrail formation, a comparative study on the onset behaviour of aircraft-generated nvPM is required."

**Other Changes**

1. Terminology: during the review we noticed that our use of terminology surrounding hygroscopicity and hydrophobicity was potentially ambiguous. We have clarified this in several instances as outlined below:

[main text: lines 15 – 16]:

"We generate lubrication oil droplets with a geometric mean mobility diameter of (100.9 ± 0.6) nm and show that these activate to form water droplets**, which**    subsequently freeze when the temperature is below ~235 K."

[main text: lines 377 – 379]:

"Measurements were performed in the temperature range 225 – 245 K and lubrication oil droplets were found to exhibit water activation under conditions consistent with that of a  **wettable**, insoluble aerosol (Petters and Kreidenweis, 2007)."

[main text: lines 393 – 395]:

"Critically, to account for the behaviour of  **non-hygroscopic** aerosol species that activate above water saturation and for  species that activate below water saturation (e.g., via PCF), the Schmidt-Appleman Criterion may need to be revised to accurately describe contrail formation."

2. Figures: all the figures had the font (axes labels, legend etc) changed from Times New Roman (serif) to Arial (sans-serif) to increase readability.

3. Additional section in the supplementary information: an additional section was added to the supplementary information outlining differences in the experimental setup when working with jet lubrication oil and citric acid aerosol. Several minor changes were then made in the main text to accommodate this new section. Please see below the details of these changes:

[supplementary information: lines 120 – 126]:

"**S4  Citric Acid Measurements**

**To generate citric acid aerosol, the Collison 3-jet nebuliser (see Fig. 2 in the main text) was replaced with a pocket nebulizer (Omron MicroAir U22) containing a 0.01 wt % solution of citric acid in Milli-Q water (Millipore Corporation). The pocket nebulizer was placed inside a container and a controlled supply of 2 Lmin$^{-1}$ filtered air was used to displace aerosol from the container into the aerosol chamber. This system was run continually to maintain a sufficient aerosol concentration within the aerosol chamber. Expansion measurements were then undertaken as outlined in Sect 4.1 of the main text and analysis was performed as described in Sect. S3.**"

[main text: Fig. 4b caption]:

[revised manuscript text omitted]